# Intra- and intergenerational costs of handicapping in the Saffron Finch (*Sicalis flaveola*), a thraupid with delayed plumage maturation

**Andrés Gabriel Palmerio**[1]*, **Carolina Isabel Miño**[2,3], **Viviana Massoni**[1,4]

**1** Departamento de Ecología, Genética y Evolución, Facultad de Ciencias Exactas y Naturales, Universidad de Buenos Aires, Ciudad Autónoma de Buenos Aires, Argentina, **2** Laboratorio de Genética Evolutiva, Facultad de Ciencias Exactas, Químicas y Naturales, Universidad Nacional de Misiones, Posadas, Misiones, Argentina, **3** Instituto de Biología Subtropical (IBS), Consejo Nacional de Investigaciones Científicas y Técnicas – CONICET/ Universidad Nacional de Misiones, Posadas, Misiones, Argentina, **4** Instituto de Ecología, Genética y Evolución de Buenos Aires (IEGEBA), CONICET, Universidad de Buenos Aires, Ciudad Autónoma de Buenos Aires, Argentina

* apalmerio@gmail.com

## Abstract

Life history theory posits negative trade-offs between current reproduction and survival and between current and future reproduction. We tested if the Saffron Finch (*Sicalis flaveola*), a sexually dichromatic thraupid with age-related plumage coloration, conforms to the expectations under this theory. Previous studies in nest-box systems have shown that both drab young males and bright-yellow older males provide similar parental care and achieve comparable reproductive success. We experimentally handicapped females and males from both age classes by clipping of three primary remiges to manipulate their parental care efforts. We compared mating pairs conformed by handicapped males of both age classes paired with non-handicapped females, handicapped females paired with either handicapped or non-handicapped males of both age classes, against control pairs (non-handicapped individuals). We assessed weight changes, feeding rates, brooding bout duration, nest temperature, and return rates for both adults and nestlings, as well as growth rate, asymptotic weight, time spent at the nest, and fledgling success. Contrary to expectations, none of the experimental individuals adjusted their feeding rates. However, we observed intra- and intergenerational costs for handicapped females mated to second-year males, including shorter brooding bouts, greater weight loss, slower offspring growth, and lighter fledglings. Thus, in our study system, lower-quality females paired with second-year males may be maximizing reproductive success under less-than-ideal circumstances.

**Data availability statement:** The data underlying the results presented in the study are available from https://doi.org/10.6084/m9.figshare.28593299.v2.

**Funding:** Funding from this study came from CONICET (11220130100342CO) (www.conicet.gov.ar) and Universidad de Buenos Aires (X-140; X-462; 20020090200160; 20020130100772) (www.uba.ar) research grants to VM. The funders didn´t play any role in the study design, data collection and analysis, decision to publish, or preparation of the manuscript.

**Competing interests:** The authors have declared that no competing interests exist.

## Introduction

The study of life histories enables the evaluation of the energy balance that birds maintain throughout their lives, as the energy invested in one activity is subtracted from the energy available for other activities [1,2]. In species with bi parental care of offspring, the fitness of both parents is jointly affected by the flexibility -or lack of it- of the response and reproductive decisions of each sex, as well as if and how they coordinate with each other. Species with bi parental care often show inter-sexual cooperation to warrant offspring survival, but conflicts in parental care behavior may also arise [3]. Both caretakers face a trade-off between current and future reproduction and try to reduce their own effort, directly affecting the balance with their partner's effort to ensure that offspring receive enough total care to survive [2]. Negotiation between the sexes could lead to partial, full or no compensation by partners, depending on the costs and benefits associated with care [4,5]. Reproductive costs can be expressed during current or future reproduction, for example, as a reduced parent' survival [6,7] or a reduced future reproductive performance [8,9]. When energetic demands are high, the caretakers may not make a proportionately greater effort, instead maintaining reproductive effort at a fairly constant rate and transferring the costs to their young rather than compromising their own survival and future reproduction [6,10].

Experimental manipulation of reproductive effort provides a direct method for exploring the relationship between energy invested in current reproduction and that allocated to future reproduction [11], thereby improving our understanding of avian reproductive behaviour [12]. In birds, a common experimental technique for testing changes in the costs of parental care involves temporarily handicapping one parent by removing flight feathers [13]. This handicap increases the energetic demands of flight by raising wing loading [14,15], with the effects being most pronounced during the nestling period, when adults experience peak energy expenditure due to food provisioning. Pioneer studies have shown that the natural daily energy expenditure of birds during reproduction ranges from 2.5 to 4 times their basal metabolic rate [16], with expenditures exceeding this range potentially having detrimental effects on future reproductive output [17]. The evolution of life-history traits associated with reproductive effort has led to a situation where individuals may risk increasing their own mortality, making variation in body mass a highly relevant factor [18]. Indeed, evidence suggests that adult mass plays a key role in foraging behaviour, food provisioning, and the regulation of parental effort in birds [19,20], and it serves as a validated index of body condition [21]. However, to more accurately estimate the relative effects across all family members—and given that variations in body mass among handicapped birds can be challenging to interpret—it is essential to directly measure parental effort through behavioural responses, such as changes in feeding rates [22], alongside offspring weight [23]. Most studies on the costs of parental care have traditionally focused on intra-individual trade-offs, while relatively few have examined the fitness consequences of parental decisions on offspring (i.e., intergenerational trade-offs; [23]).

One key variable in the early development of altricial nestlings is internal nest temperature, which is influenced by brooding behaviour. The early survival of nestlings

depends, in part, on maintaining a minimum threshold level of heat necessary for the chemical reactions that support growth [24]. Thermoregulatory behavior is energetically costly [25] and may divert time from other activities; thus, a trade-off may exist between maintaining an optimal temperature and engaging in behaviors such as foraging or caring for offspring [26].

According to the life history theory, young and inexperienced individuals are expected to prioritize their own survival over that of their offspring, whereas older individuals are more likely to invest in reproduction at the expense of their own survival in response to increased demands from nestlings [6]. In species with age-dependent phenotypes or delayed plumage maturation (reviewed in [27]), second-year males (hereafter SY males) exhibit inconspicuous plumage, often resembling that of dull females, while after-second-year males (hereafter ASY males) display more striking plumage. Although young SY males may opportunistically reproduce, they typically invest lower reproductive effort, resulting in reduced reproductive success. In contrast, older ASY males are more likely to invest in current reproductive output, even at the cost of reduced survival and body condition [27], and references therein).

Over a third of the world's cavity-nesting avian species are found in the Neotropics [28]. Thraupids account for 12% of global avian species but remain relatively understudied [29]. We focus on the Saffron Finch *Sicalis flaveola pelzelni* P. L Sclater 1872, a non-excavator obligate cavity-nesting thraupid that inhabits a large portion of northeastern and central Argentina [30]. This subspecies is sexually dichromatic and exhibits delayed plumage maturation in males; second-year (SY) males have a brown back and a whitish belly, making them indistinguishable from females to both humans and conspecifics [31], while after-second-year (ASY) males are golden yellow with an olive back streaked with black. Second-year males successfully reproduce in a nest-box system and show no significant differences in any reproductive variables when compared to ASY males. We found no differences in body size or weight between ASY and SY males; however, females mated to ASY males were significantly heavier than those mated to SY males [32]. Females contribute to nest building, incubate the eggs, and brood the nestlings alone [32]. Females Saffron Finches demonstrate strong temporal consistency in these behaviours, *sensu* [33] (i.e., the sex that invests in one type of parental care is more likely to continue investing in subsequent stages of the breeding cycle). Females also feed the offspring significantly more frequently than males [34], with this behaviour being positively correlated with the aforementioned tasks. It is worth noting that parental effort in this species is not correlated between mates [34]. Thus, Saffron Finches provide an ideal model for investigating whether individuals adjust their parental effort in challenging situations, either for themselves or to their mates.

Here, we investigate whether experimentally handicapped second-year and after-second-year males or females differ in parental effort when facing increased reproductive costs from flight, foraging, and feeding nestlings. Life-history theory predicts age-dependent responses: younger handicapped males will reduce offspring feeding to preserve their own condition, transferring costs to mates or offspring, while older males will prioritize current reproduction and absorb the handicap's costs [6]. Building on our previous studies in this species [32,34], and under this theoretical framework, we predict the following: 1) Experimentally handicapped ASY males will absorb the treatment costs and, compared to controls, will: (1.1) maintain offspring provisioning despite weight loss; (1.2) receive no compensatory care from mates but maintain nest temperature, leading to no differences in nestling development, fledging success, or recruitment; and (1.3) show reduced return rates the following season, reflecting potential long-term costs. 2) Experimentally handicapped females paired with control ASY males will absorb the costs and, compared to controls, will: (2.1) maintain similar provisioning and brooding durations but lose some body condition; (2.2) receive no compensatory paternal care but maintain optimal nest temperature, resulting in comparable nestling development, fledging success, and recruitment; and (2.3) show reduced return rates, indicating potential long-term costs; 3) Experimentally handicapped SY males will transfer costs to mates and offspring and, compared to controls, will: (3.1) reduce offspring provisioning while maintaining body condition; (3.2) receive no maternal compensation, resulting in slower nestling growth, lower asymptotic weights, longer nestling periods, reduced fledging success, and lower recruitment; and (3.3) return to the study area at similar rates, indicating comparable survival; and.4) Experimentally handicapped females paired with SY males will transfer costs to mates and offspring

and, compared to controls, will: (4.1) reduce provisioning and brooding durations while maintaining body condition; (4.2) receive no paternal compensation, leading to lower nest temperatures and nestlings with slower growth, lower asymptotic weights, longer nestling periods, reduced fledging success, and decreased recruitment; and (4.3) return to the study area at similar rates, indicating comparable adult survival.

## Materials and methods

Data for this study were collected during the 2007–2010 reproductive seasons (December to March) in an agricultural landscape within the Depressed Pampa, located in Chascomús, Buenos Aires, Argentina (35°34'S, 58°01'W). To manipulate flight ability, we clipped the 4th, 6th, and 8th primary feathers at the base of one adult in each of 54 experimental pairs. The study involved the following mating pair configurations: 1) handicapped ASY males mated to non-manipulated females (n = 15); 2) handicapped females paired with non-handicapped ASY males (n = 14); 3) handicapped SY males mated to non-manipulated females (n = 12); and 4) handicapped females paired with non-manipulated SY males (n = 13). Each experimental group had a corresponding control group (non-handicapped birds). Nests were randomly assigned to different treatments by flipping a coin at the time of first capturing an attending adult. To ensure consistent initial breeding effort across groups, we manipulated birds at nests containing 4 nestlings (the population mean, [32]). Upon capture, adults were fitted with a numbered aluminium ring and a unique combination of colored plastic rings for further identification. We captured the experimental birds on the day when the majority of eggs hatched (day 1) and re-captured them 10 days later to weigh them, by which time the nestlings had reached their asymptotic weight [32]. For each adult, we measured tarsus length using a digital caliper (± 0.01 mm) and weight with a Pesola scale (30 g ± 0.5 g). Physical condition was calculated as the ratio of weight to tarsus length. Upon re-capture (10 days after initial capture), we re-weighed the birds as a proxy for self-cost (or flexible response). The same measurements were taken for each adult in the control pairs (non-manipulated birds). The morphology and weight of individuals met the assumptions of normality and homoscedasticity, and experimental pairs were compared to control groups using ANOVA. We conducted a Pearson correlation test to assess relationships with physical condition. Finally, we compared the percentage of nest abandonment across all groups using contingency tables.

We recorded the parental behavior of adults at both experimental and control nests, on day 5 of the nestlings' age (where the growth rate is maximum), over 4 consecutive hours using Sony Hi 8 CCD-TRV 128 videocassette recorders (Sony, Tokyo, Japan), positioned 20–30 m from the nest (as described by [34]). From the videos, we registered the frequency of feeding visits by males and females (number of visits per hour) and the duration of brooding bouts by females. Females were considered to be brooding if they remained in the nest for over one minute.

We weighed the nestlings every 2 days from day 1 (hatching) to day 12 of life using Pesola scales of 10 g and 30 g (accuracy ± 0.2 g and ± 0.5 g, respectively). Each nestling was identified by marking its tarsus with a permanent marker in a different color. All measurements were taken by the same person (AGP) to minimize errors. Nestlings were not weighed after day 12 to avoid premature fledging (which typically occurs on day 14) [32]. On day 8, all nestlings were fitted with a numbered aluminium ring. We assessed the weight gain of the nestlings by fitting their growth data to a Gompertz sigmoid function ($y_{(t)} = ae^{x}$)), where $x = e^{Kt}$, $a$ represents the asymptotic weight, and $K$ is the growth rate. We averaged the variables for nestlings within a nest and compared the values of $a$ and $K$ between nestlings from manipulated and control nests.

We recorded the length of the nestling period and the proportion of fledglings. An empty nest on day 14 was assumed to indicate full successful fledging. Additionally, we registered and compared the proportion of nestlings from both experimental and control nests that returned to our study site the following year. The comparison between groups was made using contingency tables.

To investigate whether experimental and control nestlings differed in growth rate, asymptotic weight, time spent at the nest, proportion of fledglings, feeding rates, and average brooding time on day 5, we compared these variables using a

Generalized Linear Mixed Model (GLMM) with a free distribution, implemented in *R* v4.2.3 statistical environment [35]. Initially, 'Treatment' and 'Year' were included as fixed factors, while the initial physical condition of adults and the standardized laying date (by year) were included as random factors. As the effect of 'Year' was not significant in any of the analyses ($p > 0.5$), it was subsequently treated as a random factor to account for potential variation. For 'growth rate' and 'asymptotic weight' as response variables, we used a normal error function with an identity link function. For the proportion of fledglings and time spent at nests, we used a Poisson error function with a logarithmic link function. In all models, a stepwise backward elimination procedure based on Akaike's Information Criterion was employed to select the best-fitting model.

The difference in adult weight (i.e., weight on day 10 of the nestlings' life minus weight on day 1) was modeled using a GLMM with 'treatment' as a fixed predictor and 'year' and 'physical condition at the time of the first capture' as random factors, applying an identity link function. The difference in temperature was used as a proxy for brooding behavior in experimental and control nests. To assess this, ambient temperature (Ta) and nest temperature (Tn) were recorded every 3 minutes from hatching until the day after the last fledgling, using HOBO H8 data loggers with U12 sensors (precision±0.25 °C). Data for both ambient and nest temperatures were downloaded and stored on a computer using BoxCar Pro Software v3.51. The average environmental temperature and the temperature of each nest were computed by dividing the 24-hour day into four 6-hour intervals (00:00–05:59, 06:00–11:59, 12:00–17:59, and 18:00–23:59). However, as the 06:00–11:59 interval showed the greatest thermal amplitude in environmental temperature, only data for this interval was analyzed. To assess whether handicapping affected nest temperature, we fitted GLMMs with 'treatment', male age class (SY/ASY), and the sex of the handicapped bird (F/M) as fixed factors. The identity (ring number) of the females, as well as season and standardized laying date by year, were included as random factors. A normal error distribution function with an identity link function was applied. Day-slot and nestlings' age were set as cross factors of repeated measures in the model.

## Results

Ninety Saffron Finch nests were used in the experiment, divided as follows: 15 nests with manipulated ASY males, 14 nests with manipulated females paired with ASY males, 12 nests with manipulated SY males, 13 nests with manipulated females paired with SY males, 18 nests with non-manipulated ASY males and females, and 18 nests with non-manipulated SY males and females. Nest abandonment was below 10% in both manipulated and control nests (contingency tables: $\chi^2 > 0.3$, $p > 0.3$ in all cases, 3 of 54 manipulated nests and 2 of 36 control nests). We describe the results following the ontological development of the breeding cycle.

The physical condition of ASY males on day 1 of the nestlings' life (1.06±0.08 g/mm) did not differ significantly from that of SY males (1.04±0.06 g/mm, F = 1.496, p = 0.225; mean difference = 0.02 g/mm, 95% CI: −0.013–0.053). Likewise, the physical condition of females paired with ASY males (1.05±0.07 g/mm) did not differ significantly from that of females paired with SY males (1.06±0.06 g/mm, F = 0.015, p = 0.904; mean difference = −0.01 g/mm, 95% CI: −0.05–0.04). There was no significant correlation between the physical condition of handicapped birds on day 1 of the nestlings' life and that of their respective mates (S1 Table).

Feeding visits by handicapped birds, their respective partners, and non-handicapped individuals did not differ significantly (females: average feeding rate: 1.82±0.90 visits/hour, p = 0.57; males: average feeding rate: 1.50±0.11 visits/hour, p = 0.31; mean difference = 0.32 visits/hour, 95% CI: −0.79–1.43). However, brooding bouts of handicapped females paired with non-handicapped SY males (9.2±3.1 min, 95% CI 3.1–15.3 min) were significantly shorter compared to those of non-handicapped females mated to non-handicapped SY males (controls) (13.9±2.9 min, 95% CI 8.2–19.6 min, Wald's = 2.59, F = 6.21, p = 0.01, mean difference = −4.7 min, 95% CI: −8.2–1.2 min. Fig 1, Table 1). In addition, the change in weight (from d 1 to d 10 of nestlings' life) of handicapped females paired with control SY males differed significantly from that of control females (Fig 2, S2 Table). These experimental females weighed 0.75 g less than their non-handicapped counterparts, representing a 7.5% reduction in weight compared to control SY females.

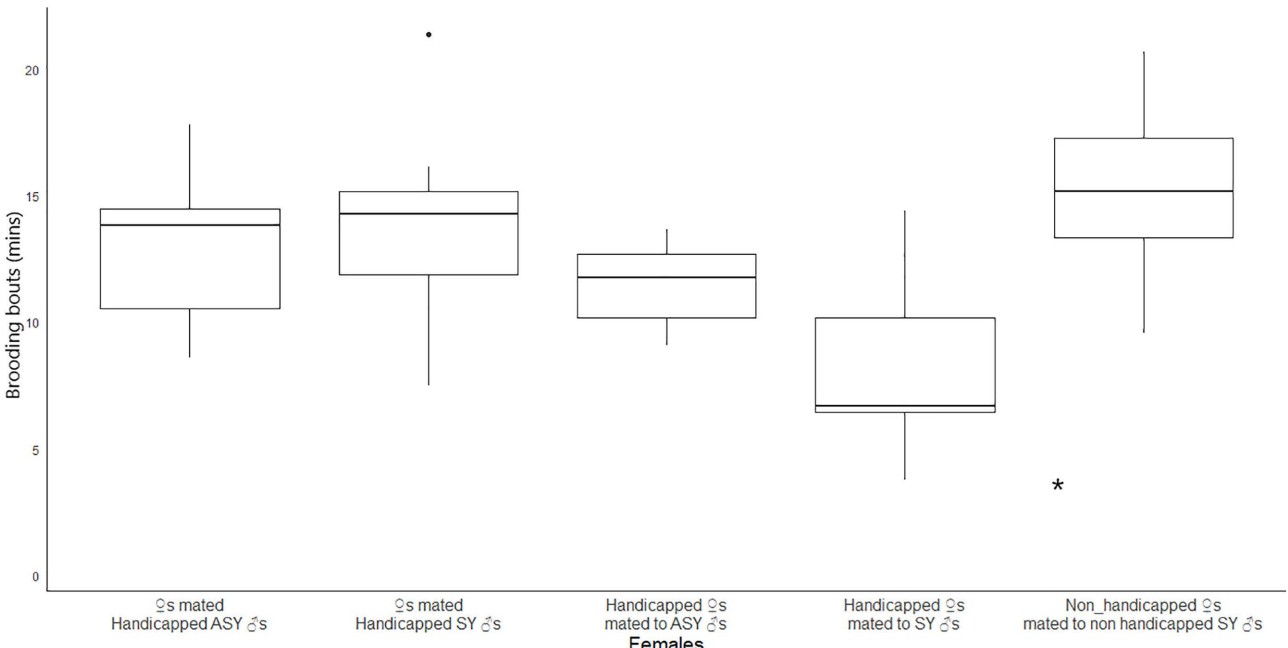

**Fig 1. Box plots of average brooding bouts of females at experimental and control nests.** The asterisk indicates a significant difference (p<0.05).

**Table 1. Effects of experimental handicapping costs on feeding rates at day 5 of nestlings' life in Saffron Finches. Results of Generalized Linear Mixed Models testing the effects of experimental handicapping on feeding rates (number of visits/hour), and on the average duration of brooding behaviour (min). The mean value of each variable ± standard errors (S.E.) and 95% confidence interval (95% CI) are shown.** *W*: Wald's statistic. The number in bold font (last line) indicates a significant *p*-value (< 0.05).

| Comparison | Sex | Feeding rate | | | | | Brooding length | | | | |
|---|---|---|---|---|---|---|---|---|---|---|---|
| | | *W* | *F* | *p-value* | Mean ± *SE* | 95% CI | *W* | *F* | *p-value* | Mean ± *SE* | 95% CI |
| **Handicapped ASY males vs. control** | Males | 4.06 | 0.38 | 0.67 | 1.4 ± 0.2 | −0.7-2.3 | 0.83 | 0.21 | 0.38 | 10.7 ± 3.2 | 4.5-16.9 |
| | Females | 5.02 | 0.20 | 0.60 | 1.9 ± 0.3 | −0.3-3.9 | 0.20 | 1.43 | 0.66 | 12.5 ± 2.1 | 8.4-16.6 |
| **Handicapped females mated to ASY males vs. control** | Males | 2.67 | 0.19 | 0.29 | 1.6 ± 0.3 | −0.5-2.6 | 4.02 | 2.68 | 0.19 | 11.7 ± 3.2 | 5.5- 17.9 |
| | Females | 6.32 | 2.16 | 0.32 | 1.7 ± 0.6 | −0.9-3.1 | 2.59 | 5.52 | 0.26 | 11.7 ± 2.9 | 6.0-17.4 |
| **Handicapped SY males vs. control** | Males | 0.08 | 1.23 | 0.70 | 1.3 ± 0.7 | −1.2-2.2 | 0.85 | 2.01 | 0.93 | 12.7 ± 4.1 | 4.7-20.7 |
| | Females | 1.36 | 1.40 | 0.25 | 2.1 ± 0.4 | −0.2-3.3 | 0.30 | 1.56 | 0.24 | 16.4 ± 3.3 | 10.0-22.8 |
| **Handicapped females mated to SY males vs. control** | Males | 1.21 | 0.42 | 0.55 | 1.4 ± 0.2 | −0.6-2.7 | 5.38 | 0.22 | 0.22 | 10.3 ± 3.9 | 2.6-18.0 |
| | Females | 0.56 | 0.73 | 0.49 | 2.0 ± 0.8 | −0.3-3.1 | 2.59 | 6.21 | **0.01** | Handicapped 9.2 ± 3.1 | Handicapped 3.2–15.2 |
| | | | | | | | | | | Control 13.9 ± 2.9 | Control 8.2–19.6 |

The growth rate and asymptotic weight of nestlings born in most of experimentally manipulated nests did not differ significantly from those of nestlings born in control nests, with the exception of nestlings born in nests attended by handicapped females mated with SY males (Table 2, Fig 3). These nestlings grew 20% slower and weighed 22% less than those born in control nests.

Nest temperature did not differ significantly between experimental and control nests (p>0.1 in all cases, S3 Table). However, the temperature in experimental nests attended by handicapped females paired with SY males was consistently lower

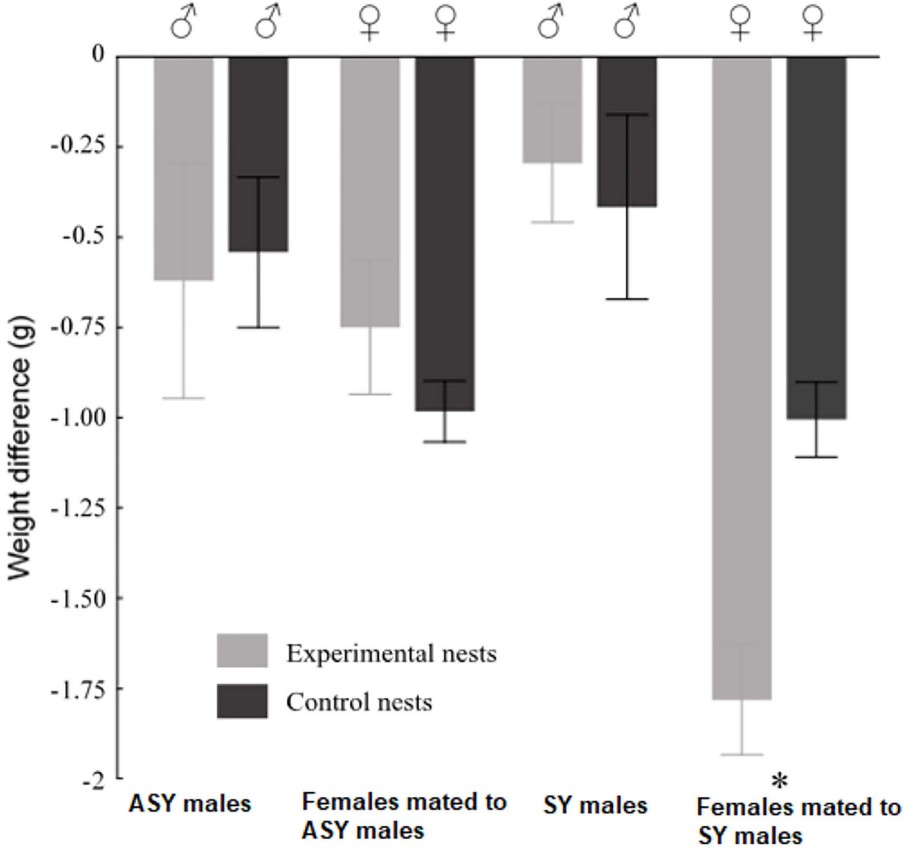

**Fig 2. Average difference in weight of adults attending experimental and control nests.** The mean value of the difference in adults' weight at day 10 minus adults' weight at day 1 of nestlings' life ± standard errors (S.E.) are reported. The asterisk indicates a significant difference (p < 0.05).

**Table 2. Effect of experimental handicapping of adults on the growth rates and average asymptotic weight of nestlings in Saffron Finches. Results of the Generalized Linear Mixed Model testing the effects of handicapping of adults on the growth rates and average asymptotic weight of nestlings born at experimental and control nests. The mean value of each variable ± standard errors (S.E.) and 95% confidence interval (95% CI) are shown. W: Wald's statistic. The numbers in bold font indicate significant p-values (< 0.05).**

| Comparison | Growth rate | | | | | Asymptotic weight | | | | |
|---|---|---|---|---|---|---|---|---|---|---|
| | W | F | p-value | Mean (K) ± SE | 95%CI (K) | W | F | p-value | Mean (g) ± SE | 95% CI (g) |
| **Handicapped ASY males vs. control** | 0.15 | 1.41 | 0.31 | 0.31±0.12 | 0.07-0.55 | 1.23 | 1.88 | 0.39 | 14.6±3.2 | 8.4-20.8 |
| **Handicapped females mated to ASY males vs. control** | 1.87 | 0.46 | 0.76 | 0.30±0.18 | −0.06-0.66 | 0.60 | 0.33 | 0.57 | 14.4±4.2 | 6.2-22.6 |
| **Handicapped SY males vs. control** | 0.05 | 0.05 | 0.83 | 0.33±0.11 | 0.11-0.55 | 0.10 | 1.23 | 0.23 | 14.8±3.5 | 7.9-21.7 |
| **Handicapped females mated to SY males vs. control** | 3.77 | 0.86 | **0.02** | Handicapped 0.26±0.09 | Handicapped 0.09-0.43 | 2.88 | 0.36 | **0.01** | Handicapped 13.1±2.9 | Handicapped 7.4–18.8 |
| | | | | Control 0.31±0.09 | Control 0.14-0.48 | | | | Control 14.8±3.1 | Control 8.7-20.9 |

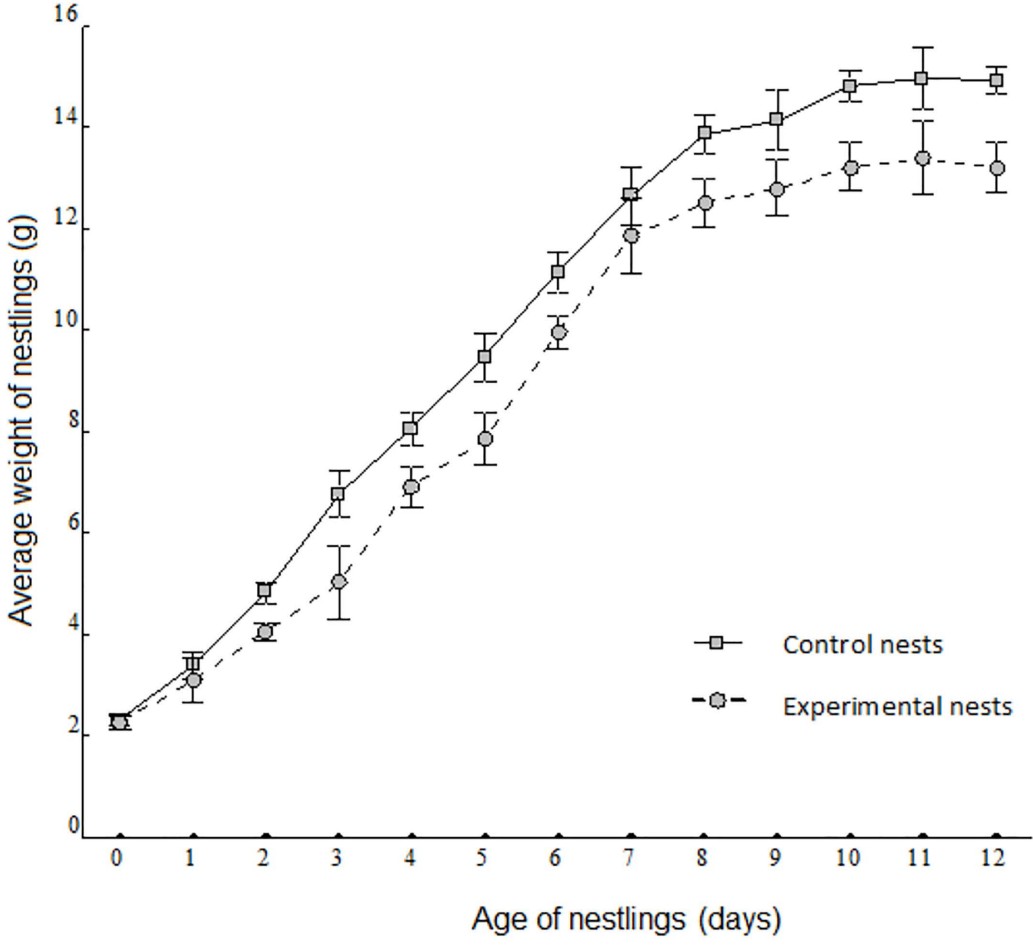

**Fig 3. Average weight of Saffron Finch nestlings born in nests attended by experimental and control nests.** Plot of average weight of nestlings born in nests attended by handicapped females mated with non-handicapped SY males (experimental nests, dashed line), and of nestlings born in nests attended by non-manipulated birds (control nests, solid line).

than that of their respective control nests throughout the entire nestling period (Fig 4). The nestlings remained in the nest for a similar amount of time and fledged at comparable rates, irrespective of the treatment (p>0.05 in all cases, Table 3).

Likewise, the return rates of nestlings born in experimental nests did not differ from those born in control nests (*p*>0.05 in all cases, Table 4). The return rates of handicapped adults were also similar to those of non-manipulated adults (*p*>0.05 in all cases, Table 4).

## Discussion

This experimental study in the Saffron Finch evaluated whether handicapping of males of different age classes and handicapping of the females mated to those males affected parental care. None of the experimental individuals adjusted the feeding rates of nestlings, but, notably, the females mated to non-handicapped SY males, as well as their offspring, bore the costs of the experiment, in different ways.

Saffron Finch males face strong sexual selection – they display sexual dichromatism and carotenoid-based coloration – and paternity uncertainty [36]. Such species seem to show a trend towards female-biased parental care [33] (and references therein). Paternity uncertainty selects against male care in birds [37]. Indeed, extra-pair paternity was shown to be

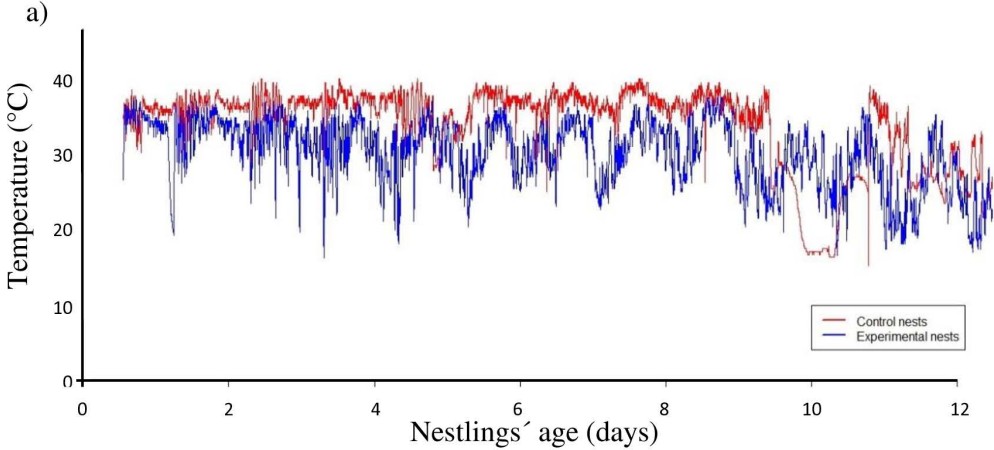

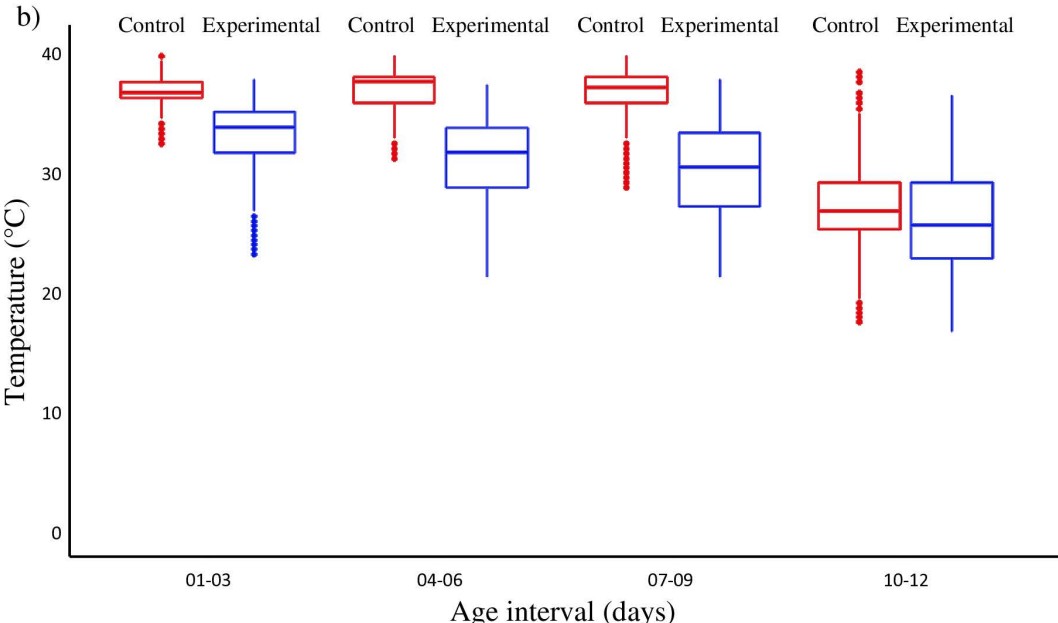

**Fig 4. Handicapping costs and nest temperature in Saffron Finches.** (a) Plot of nest temperature along the nestling period in nests attended by control nests (red) and by handicapped females mated to SY males (blue); (b) Box-plots of average temperature (± *S.E.*) of nests attended by control birds (red) compared with experimental nests attended by handicapped females mated to SY males (blue).

associated with reduced male care across different parental care forms in birds [33] (and references therein). In Saffron Finches, ASY males gain extra-pair paternity from SY males or from other ASY males, mediated by differences in song quality, another age-related trait [38]. The bright yellow plumage of ASY male Saffron Finches comes from lutein, a carotenoid acquired through diet [38]. A multispecies research found a negative association between carotenoid-dependent plumage coloration and parental care in passerines [39]. Ongoing work in Saffron Finches is looking at differences in the orange forehead of ASY males, which is a dynamic character that appears only during the breeding season, and its relationship with the amount of parental care provided.

**Table 3. Effects of handicapping costs of Saffron Finches parents on the time spent by nestlings at nests and on the proportion of fledglings.** Results of the Generalized Linear Mixed Model testing for effects of handicapping of males or females on the time spent by nestlings at nest and on the proportion of fledglings. *W*: Wald's statistic. Mean values ± Standard Errors (*SE*) and 95% confidence interval (95% CI) are shown for each variable.

| Comparison | Age of nestlings (days) | | | | | Percentage of fledglings | | | | |
|---|---|---|---|---|---|---|---|---|---|---|
| | *W* | *F* | *p-value* | Mean ± *SE* | 95%CI | *W* | *F* | *p-value* | Mean ± *SE* | 95%CI |
| **Handicapped ASY males vs. control** | 0.17 | 0.08 | 0.92 | 14.3 ± 1.3 | 11.7-16.9 | 0.05 | 0.34 | 0.82 | 87 ± 5.2 | 76.8-97.2 |
| **Handicapped females mated to ASY males vs. control** | 0.74 | 0.37 | 0.49 | 14.3 ± 1.1 | 12.1-16.5 | 0.19 | 0.20 | 0.67 | 86 ± 8.8 | 68.7-103.3 |
| **Handicapped SY males vs. control** | 0.71 | 0.35 | 0.72 | 14.4 ± 0.9 | 12.6-16.2 | 0.38 | 0.11 | 0.56 | 89 ± 7.6 | 74.1-103.9 |
| **Handicapped females mated to SY males vs. control** | 0.52 | 0.34 | 0.36 | 14.1 ± 0.8 | 12.5-15.7 | 0.66 | 0.39 | 0.83 | 84 ± 7.1 | 70.1-97.9 |

**Table 4. Handicapping costs and return rates.** Results of the contingency table test evaluating if handicapping of males or females affects the return rates of nestlings and adults to the study site. The number of comparisons made and the percentage (%) of returning nestlings and adults are also shown.

| Comparison | Return rate of nestlings | | | | Return rate of adults | | | |
|---|---|---|---|---|---|---|---|---|
| | Chi-square | *p-value* | *n* | % | Chi-square | *p-value* | *n* | % |
| Handicapped ASY males vs. control | 0.34 | 0.24 | 33 | 1.8 | 0.17 | 0.76 | 33 | 35 |
| Handicapped females mated to ASY males vs. control | 0.49 | 0.76 | 32 | 1.7 | 0.81 | 0.42 | 32 | 40 |
| Handicapped SY males vs. control | 0.63 | 0.46 | 30 | 2.1 | 1.14 | 0.23 | 30 | 33 |
| Handicapped females mated to SY males vs. control | 0.77 | 0.19 | 31 | 1.5 | 0.93 | 0.38 | 31 | 38 |

Theoretical assumptions posit that older males would favour reproduction over survival [40]. In line with such expectations, we found that more experienced ASY males, even handicapped, as well as the females mated to non-handicapped ASY males, maintained expected feeding rates (Table 1), and raised their offspring at the expected growth rate and reaching the expected asymptotic weight (Table 2). Neither males nor females lost significant weight in the process when compared to control nests (Fig 2), suggesting that they overcame the experimentally increased reproduction cost. Contrary to predictions, however, inexperienced and handicapped SY males also fed their offspring at the expected rate (Table 1), did not lose significant weight (Fig 2), and their offspring developed equally to those of control nests (Table 3). Instead, handicapped females mated to SY males also maintained the feeding rate, reducing the length of brooding bouts, at the expense of significantly losing weight (intra-generational cost of the experiment, Table 1), and their offspring grew at a slower pace and were lighter than the nestlings in the control nests (inter-generational cost of the experiment, Fig 3).

Our results in the Saffron Finch agree to the prediction that the mates of handicapped individuals would not adjust their behaviour to compensate costs, in line with evidence from our previous study in this species showing independence of parental behavior between mates [34]. Our findings in Saffron Finches are compatible with the flexible investment hypothesis [41], which suggests that birds adjust their reproductive effort to the demands of the offspring, depending on the availability of resources. With sufficient food, parents can meet the needs of their nestlings; however, this compensation is not possible when resources are limited [42,43]. The shorter brooding bouts (Table 1) and weight loss (along the experiment, 7.5%). of experimentally handicapped females mated to SY males (Fig 2) suggest that they were unable to forage properly, and provided care at their maximum capacity [33]. Similar findings have been reported in the Great Tit *Parus major* [12] and in the Pied Flycatcher *Ficedula hypoleuca* [44]. Thus, Saffron Finch SY males, which presumably are of an intrinsic low quality, might be doing the best of a bad job, when mated to those females. Indeed, experimentally handicapped females mated to SY males substantially maintained the optimal feeding rate by reducing the duration of the brooding bouts (Table 1). Our findings do not conform to the 'partial compensation' or the 'partner unawareness of mate's condition' explanations [45] because Saffron Finches handicapped females reduced brooding bouts, and this is a behavioral trait that males cannot compensate, even if they were aware of it, because this sex does not incubate.

Interestingly, the temperature of experimental nests did not change (Figs 4a and 4b). We can speculate that part of the energy received by the offspring from females in the form of food, probably went into re-heating their bodies before being diverted for growth, which requires an average body temperature of 39 Cº. This would explain the lower growth rate and asymptotic weight showed by those nestlings (Table 2). On the other hand, probably, the cavity-type nest prevented the nestlings from losing heat beyond the minimum threshold required for growth [46], allowing them to continue growing, albeit at a slower pace (Fig 3). A reduction in weight despite maintaining the same feeding rate to offspring, other than inability to self-forage could be that handicapped females may experience inefficient and higher costs of flying performance due to handicapping. Further studies with a larger sample size might confirm significance of the trends observed here.

We also found that Saffron Finches show high temporal consistency in female parental care. Previous multispecies studies, did not consider brooding as a parental care variable [33]. Our study assessed brooding efforts because this period is likely the most energy-intensive of the nestling cycle, as nestlings require both warmth and food at an increasing rate. In species with sex-biased parental care, the brooding sex would have to bear an even higher demand, foraging for themselves and for offspring. Future studies should focus more on this relatively underexplored aspect of parental care (but see [22,26,47]).

Fledging success is a reliable index of recruitment [48]. Indeed, a study on 65 altricial bird species found that individual survival depends strongly on physical traits such as mass [49]. We found that Saffron finch nestlings born in experimental and in control nests spent a comparable amount of time in the nest before fledging and fledge in a similar proportion (Table 3). From this, and evidence discussed above, we can infer that the offspring of handicapped females mated to SY males fledged weighing less than offspring of control pairs. Being lighter at fledging could compromise the post-fledging survival, as in juvenile Great Tits *Parus major*, and Coal Tits *Parus ater* [50]. In addition, being lighter at fledging could affect the recruitment rate [51]. However, the recruitment rate of nestlings born at the different nest treatments (1.5 to 2.1%, Table 4) fell within the range observed for several passerines [52]. Moreover, the recruitment of Saffron Finch nestlings from nests attended by handicapped mates of SY males was similar to that of control pairs (Table 4). The overall recruitment rate of nestlings found in the present study was lower than the reported in a recent work from our group in Saffron Finches (5%, [53]), but a limited sample size prevents further explanations. The recruitment rates of experimental adults were similar to those of control adults of both sexes. The swift regeneration of primary feathers and prolonged time of re-sighting (one year) might have diluted the cost imposed to the flight of experimental individuals.

In sum, our results regarding the age of reproduction of Saffron Finch males in a nest-box system are not entirely consistent with predictions of life history theory. Evidence from this study adds up to recent studies on costs of parental care suggesting that the trade-offs between parental effort and survival are more complex than previously thought [23,40], and that life history pace and traits may play a not so significant role than previously suggested. In this sense, our findings are also in line with Williams [54] who suggests that we should not expect to see costs of reproduction associated with parental care except in low quality food stressed individuals, as could be the case of handicapped females mated to SY males. More studies are needed – especially in the many avian species with sexual selection on males and paternal uncertainty inhabiting subtropical latitudes – to help discerning the relative importance of life history traits and temporal consistency of female-biased parental care.

## Supporting information

**S1 Table. Lack of correlations in physical condition between mates, in experimentally handicapped pairs and control pairs of Saffron Finches.**
(DOCX)

**S2 Table. Average weight difference of adult females and males Saffron Finches attending experimental and control nests (weight at day 10 minus weight at day 1 of nestlings' life [± standard error (*S.E.*) and 95% Confidence Interval (CI)].**
(DOCX)

**S3 Table. Results of the Generalized Linear Mixed Model comparing the nest temperature in experimental and control nests, for all treatments.**
(DOCX)

## Acknowledgments

We are grateful to the authorities and personnel of the Instituto Tecnológico de Chascomús (INTECH-CONICET) for granting access to work and reside in the premises.

## Author contributions

**Conceptualization:** Viviana Massoni.

**Data curation:** Andrés Gabriel Palmerio.

**Formal analysis:** Andrés Gabriel Palmerio.

**Investigation:** Andrés Gabriel Palmerio, Viviana Massoni.

**Methodology:** Andrés Gabriel Palmerio.

**Supervision:** Viviana Massoni.

**Writing – original draft:** Andrés Gabriel Palmerio, Carolina Isabel Miño, Viviana Massoni.

**Writing – review & editing:** Carolina Isabel Miño.

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
