## [Decision Letter · Decision Letter 0]

23 Apr 2025

PONE-D-25-13670Intra- and intergenerational costs of handicapping in the Saffron Finch (Sicalis flaveola), a thraupid with delayed plumage maturation.PLOS ONE

Dear Dr. Palmerio,

Thank you for submitting your manuscript to PLOS ONE. After careful consideration, we feel that it has merit but does not fully meet PLOS ONE’s publication criteria as it currently stands. Therefore, we invite you to submit a revised version of the manuscript that addresses the points raised during the review process.

We look forward to receiving your revised manuscript.

Kind regards,

Shoko Sugasawa

Academic Editor

PLOS ONE

Additional Editor Comments:

Both reviewers agree that the question this study is asking is important and interesting, however, Reviewer 2 un particular raises a couple of significant points that should be addressed in text and analyses. Please read and respond to both reviewers' feedback carefully.

Reviewers' comments:

Reviewer's Responses to Questions

**Comments to the Author**

1. Is the manuscript technically sound, and do the data support the conclusions?

Reviewer #1: Yes

Reviewer #2: No

2. Has the statistical analysis been performed appropriately and rigorously? 

Reviewer #1: Yes

Reviewer #2: Yes

3. Have the authors made all data underlying the findings in their manuscript fully available?

Reviewer #1: Yes

Reviewer #2: No

4. Is the manuscript presented in an intelligible fashion and written in standard English?

Reviewer #1: Yes

Reviewer #2: Yes

5. Review Comments to the Author

Reviewer #1: Title: Intra- and intergenerational costs of handicapping in the Saffron Finch (Sicalis flaveola), a thraupid with delayed plumage maturation.

In this interesting paper, the authors performed a classical wing-clipping handicapping experiment on saffron finch parents to evaluate possible compensatory mechanisms in response to an experimental increase in parental effort by one parent. Using a well-balanced design, the authors found no behavioral responses in terms of feeding rates adjustment by the handicapped individual or its partner. However, brooding behavior differed in wing-clipped females in relation to male age. This reduction of brooding behavior could then explain the impaired thermoregulation inside the nest, and the reduced growth trajectories of the offspring.

This manuscript is overall well-written. Hypothesis and predictions are easy to understand. Methodology and statical analyses are appropriate, and the discussion is overall well laid out. I have no major concern, but I have few minor revisions to propose.

Specific revisions:

L143: I appreciate how the authors laid out all the hypotheses 1), 2), 3) and 4) and I agree with the authors about their predictions. The prediction 3.2) is however unclear/uncertain to me. Handicapped 2Y males are expected to reduce feeding rates due to transfer of handicapping costs to partner or offspring and this prediction makes sense. But why are you in this case not predicting that females will compensate? Studies have found a wide range of responses, is there a particular reason why in your study system you are specifically predicting no response?

In the prediction 4.2) I would agree with you as males typically are less responsive than females.

L187: do males also brood? If not this sentence is better rewritten as : Females were considered to be brooding is they remained in the …

L252: I think it would be nice to also show a figure about the reduction in brooding bouts in the female handicap group.

L303: no need to refer again to tables and figures in the first paragraph of the discussion.

L317: it is unclear here which findings in this study are in line with this description of male extra pair paternity. The author should clarify.

L350: a reduction in weight by handicapped females despite maintaining the same feeding rate to offspring, other than inability to self-forage as the author mentioned, could be cause by inefficient and higher costs of flying performance due to handicapping.

Reviewer #2: The manuscript, "Intra- and intergenerational costs of handicapping in the Saffron Finch (Sicalis flaveola), a thraupid with delayed plumage maturation," addresses an important question about life-history trade-offs and parental investment strategies in a Neotropical passerine with delayed plumage maturation. In many ways, this manuscript is interesting, and I do like the idea underlying it. The experimental approach is appropriate, the topic is timely, and the manuscript is structured and well-written overall.

However, I believe that the design of the study has some conceptual mistakes that decrease the robustness of the results, and this unfortunately, prevents me from being more supportive. Several major methodological and conceptual issues substantially affect the strength and interpretation of the findings. Some concerns relate to experimental design (e.g., control group handling), some to the statistical treatment of key variables (e.g., year effects), and others to biological assumptions (e.g., age categorization).

Major concerns:

1. Age categorization of ASY males. The authors pool all after-second-year (ASY) birds together, a group that likely spans individuals from 2-3 to the maximum lifespan for this species, which can reach 10-12 (in captivity). This categorization is too coarse and likely obscures important variation in reproductive strategies, as reproductive investment, body condition, and risk-taking behaviors can vary markedly with age.

2. Unmanipulated and control birds. Usually, in wing-clipping experiments, it is standard practice to handle control birds similarly to experimental birds to account for capture/handling stress, which can independently affect parental behavior. Here, control birds seem unmanipulated, raising concerns that differences attributed to handicapping might instead reflect handling effects.

3. The effect of the year in the statistical analyses. While laying dates were standardized within years, the year itself was only included as a random effect without thorough exploration. Annual variation in food availability, weather, and predation can strongly influence reproductive behaviors and nestling growth. I would include the year as a fixed factor in key analyses, test for significant year effects, or explicitly discuss that environmental variation across years could confound results.

4. The hypotheses. The predictions are laid out in a lengthy, detailed list that is difficult to follow. While specificity is good, the structure could be improved for readability.

5. Effect sizes, confidence intervals, and exact p-values are not consistently reported. Current standards favor detailed reporting to allow readers to assess the biological importance of results, even when not statistically significant.

6. The Discussion emphasizes the flexible investment hypothesis but does not adequately consider alternative explanations (e.g., constrained compensation due to energetic limits, partner unawareness of mate condition).

Minor comments:

- Line 21: replace "conform" by "conforms"

- Line 65: find a synonym for “seminal”

- Line 160: I would place the sample size for each group here. The information about the yearly sample size is missing.

- L161: What is the difference between non-manipulated birds and control birds?

- L204: I see a problem when authors compare experimental and control nestlings. Are the nestlings from unmanipulated and control pairs pooled together?

- L238: Please specify how many nests were deserted and the groups which they belonged

- Line 305: "bared" → "bore" (past tense).

- Make figure legends fully self-contained (state which line represents which group).

- Double-check minor inconsistencies in reference

6. PLOS authors have the option to publish the peer review history of their article (what does this mean? ). If published, this will include your full peer review and any attached files.

**Do you want your identity to be public for this peer review?** For information about this choice, including consent withdrawal, please see our Privacy Policy .

Reviewer #1: **Yes: ** Davide Baldan

Reviewer #2: No

---

## [Author Response · Author response to Decision Letter 1]

6 Jun 2025

Authors’ replies to reviewers comments

We are grateful for the comments and suggestions made by both reviewers. Below, we provide our replies (marked as AR, Authors’ Reply) to each of the specific comments. Line numbers refer to the ones in the track-changes version).

In addition, in reply to the reviewers’ comments, we added a new Supplementary Table (now S2 Table), we renumbered Figures and Tables and ameliorated table captions and figure legends to improve clarity.

Authors’ replies to Reviewer #1

Reviewer #1:

L143: I appreciate how the authors laid out all the hypotheses 1), 2), 3) and 4) and I agree with the authors about their predictions. The prediction 3.2) is however unclear/uncertain to me. Handicapped 2Y males are expected to reduce feeding rates due to transfer of handicapping costs to partner or offspring and this prediction makes sense. But why are you in this case not predicting that females will compensate? Studies have found a wide range of responses, is there a particular reason why in your study system you are specifically predicting no response? In the prediction 4.2) I would agree with you as males typically are less responsive than females.

AR: In a previous study on parental care in this species (Palmerio and Massoni 2011), we did not find evidence that feeding rates were correlated (either positively or negatively) between males and females, so that we do not expect that females mated to handicapped SY males will compensate.

L187: do males also brood? If not this sentence is better rewritten as “Females were considered to be brooding if they remained in the …”

AR: Thank you for noting this. The sentence was rewritten as suggested.

L252: I think it would be nice to also show a figure about the reduction in brooding bouts in the female handicap group.

AR: Thank you for this suggestion. We added a figure showing the plot of brooding bouts by females in each experimental and control group (new figure 1).

L303: no need to refer again to tables and figures in the first paragraph of the discussion.

AR: We removed references to figures and table in the first paragraph, as suggested.

L317: it is unclear here which findings in this study are in line with this description of male extra pair paternity. The author should clarify.

AR: We re-phrased, for better clarity (now lines 323-325).

L350: a reduction in weight by handicapped females despite maintaining the same feeding rate to offspring, other than inability to self-forage as the author mentioned, could be cause by inefficient and higher costs of flying performance due to handicapping.

AR: Thank you for this valuable suggestion, which we added to our discussion (lines 374-377).

Authors’ replies to Reviewer #2

We are grateful to the reviewer for the comments and suggestions. Below, we provide our replies (marked as AR, Author Reply) to each of his/her specific comments.

Reviewer #2: The manuscript, "Intra- and intergenerational costs of handicapping in the Saffron Finch (Sicalis flaveola), a thraupid with delayed plumage maturation," addresses an important question about life-history trade-offs and parental investment strategies in a Neotropical passerine with delayed plumage maturation. In many ways, this manuscript is interesting, and I do like the idea underlying it. The experimental approach is appropriate, the topic is timely, and the manuscript is structured and well-written overall.

However, I believe that the design of the study has some conceptual mistakes that decrease the robustness of the results, and this unfortunately, prevents me from being more supportive. Several major methodological and conceptual issues substantially affect the strength and interpretation of the findings. Some concerns relate to experimental design (e.g., control group handling), some to the statistical treatment of key variables (e.g., year effects), and others to biological assumptions (e.g., age categorization).

Major concerns:

1. Age categorization of ASY males. The authors pool all after-second-year (ASY) birds together, a group that likely spans individuals from 2-3 to the maximum lifespan for this species, which can reach 10-12 (in captivity). This categorization is too coarse and likely obscures important variation in reproductive strategies, as reproductive investment, body condition, and risk-taking behaviors can vary markedly with age.

AR: Average annual survival rates for Saffron Finches suggest a typical lifespan of 2–3 years in the wild (Palmerio and Massoni 2024). In Saffron Finches, as in many other species showing delayed plumage maturation (DPM, Hawkins 2012), the difference in plumage morphs between second-year and after second-year individuals is used as a proxy for age-class categorization (Hawkins 2012, Cimprich 2019, Pruett et al. 2017, De Lima et al. 2025, Morosse et al. 2025).

In addition, longevity of captive birds is not representative of longevity in nature, because most often such birds are free of the many threats to survival faced by wild birds, such as habitat change, harsh weather, low availability of insects, incidental loss, competition, predation and disease exposure (Imlay & Leonard 2019).

References cited in this reply:

Hawkins GL, Hill GE, Mercadante A. 2012. Delayed plumage maturation and delayed reproductive investment in birds. Biol Rev Camb Philos Soc. 87(2):257-74. doi: 10.1111/j.1469-185X.2011.00193.x.

Cimprich, DA. 2019. An Evaluation of Characters for Age and Sex Determination of the Blackcapped Vireo. North American Bird Bander 44: 149-159.

Pruett HL, Long HM, Mathewson HA, Morrison ML. 2017. Does Age Structure Influence Golden-Cheeked Warbler Responses Across Areas of High and Low Density? Western North American Naturalist 77(4): 421-429.

Morosse OJ, Tsunekage T, Kenny-Duddela HV, Schield DR, Keller KP, Safran RJ, Levin I. 2025. North American barn swallows pair, mate, and interact assortatively, Behavioral Ecology, https://doi.org/10.1093/beheco/araf060

Imlay TL & Leonard ML. 2019. A review of the threats to adult survival for swallows (Family: Hirundinidae). Bird Study, 66(2): 251-263.

2. Unmanipulated and control birds. Usually, in wing-clipping experiments, it is standard practice to handle control birds similarly to experimental birds to account for capture/handling stress, which can independently affect parental behavior. Here, control birds seem unmanipulated, raising concerns that differences attributed to handicapping might instead reflect handling effects.

AR: Control birds were handled similarly to experimental birds: we captured them, handled them and took the same measurements than in experimental birds (mentioned at lines 175-176 of our original submission).

3. The effect of the year in the statistical analyses. While laying dates were standardized within years, the year itself was only included as a random effect without thorough exploration. Annual variation in food availability, weather, and predation can strongly influence reproductive behaviors and nestling growth. I would include the year as a fixed factor in key analyses, test for significant year effects, or explicitly discuss that environmental variation across years could confound results.

AR: Thank you for this suggestion. We have included ‘Year’ as a fixed factor in GLMMs analyses, but results proved not significant (see lines 201-205). We also tested for the effect of year on the response variables examined, but results proved non-significant (see additional files for the reviewers)

4. The hypotheses. The predictions are laid out in a lengthy, detailed list that is difficult to follow. While specificity is good, the structure could be improved for readability.

AR: We re-wrote the predictions, maintaining specificity, but improving the structure for readability. We hope that this new version is easier to follow (lines 119-147).

5. Effect sizes, confidence intervals, and exact p-values are not consistently reported. Current standards favour detailed reporting to allow readers to assess the biological importance of results, even when not statistically significant.

AR: Thank you. The effect sizes, confidence intervals, and exact p-values are now reported in the tables, in compliance with current standards.

6. The Discussion emphasizes the flexible investment hypothesis but does not adequately consider alternative explanations (e.g., constrained compensation due to energetic limits, partner unawareness of mate condition).

AR: Thank you for your suggestion. In the discussion, as suggested by the reviewer, we now also comment on possible alternative explanations: the constrained compensation due to energetic limits and the partner unawareness of mate condition (lines 365-377).

The constrained compensation hypothesis addresses how parents adjust their care when their partner reduces effort, particularly under energetic limitations. It states that when one parent reduces care, the other can only partially compensate due to physiological or energetic constraints, resulting in less-than-complete replacement of lost parental effort. This hypothesis predicts that when one parent reduces care, the other parent only partially compensates. However, in our study, the females did not reduce feeding rates, they reduced brooding bouts, but given that males do not brood, they cannot compensate this aspect of parental care.

The partner unawareness of mate condition hypothesis addresses how parents adjust their caregiving based on their own condition versus their partner’s condition. In the context of parental care, this hypothesis states that one parent adjusts their level of care according to their own physical or energetic state, but does not respond to changes in their partner’s condition. In our study, the sexes did not adjust the feeding rates of nestlings. Handicapped females reduced their brooding bouts, but their partners cannot respond to this change – even if they were aware of it – because Saffron Finch males do not brood.

R#2 Minor comments:

- Line 21: replace "conform" by "conforms"

AR: replaced, as indicated.

- Line 65: find a synonym for “seminal”

AR: Thank you for this suggestion. Changed to “pioneer” studies.

- Line 160: I would place the sample size for each group here. The information about the yearly sample size is missing.

AR: We have added sample sizes for each group. Thank you!

- L161: What is the difference between non-manipulated birds and control birds?

AR: There is no difference; we took non-manipulated birds as “control birds”.

- L204: I see a problem when authors compare experimental and control nestlings. Are the nestlings from unmanipulated and control pairs pooled together?

AR: Nestlings from unmanipulated and control pairs are the same nestlings (unmanipulated are control pairs).

- L238: Please specify how many nests were deserted and the groups which they belonged

AR: We now specify the number of deserted nests and the groups to which they belonged (lines 232-235).

- Line 305: "bared" → "bore" (past tense).

AR: Thank you! Corrected, as indicated.

- Make figure legends fully self-contained (state which line represents which group).

AR: Thank you. Figure legends have been edited to make them fully self-contained.

- Double-check minor inconsistencies in reference

AR: Thank you. We have made our best to double-check and corrected the references.

---

## [Decision Letter · Decision Letter 1]

13 Aug 2025

Intra- and intergenerational costs of handicapping in the Saffron Finch (Sicalis flaveola), a thraupid with delayed plumage maturation.

PONE-D-25-13670R1

Dear Dr. Palmerio,

We’re pleased to inform you that your manuscript has been judged scientifically suitable for publication and will be formally accepted for publication once it meets all outstanding technical requirements.

Kind regards,

Shoko Sugasawa

Academic Editor

PLOS ONE

Reviewers' comments:

Reviewer's Responses to Questions

**Comments to the Author**

1. If the authors have adequately addressed your comments raised in a previous round of review and you feel that this manuscript is now acceptable for publication, you may indicate that here to bypass the “Comments to the Author” section, enter your conflict of interest statement in the “Confidential to Editor” section, and submit your "Accept" recommendation.

Reviewer #1: All comments have been addressed

Reviewer #2: All comments have been addressed

2. Is the manuscript technically sound, and do the data support the conclusions?

Reviewer #1: Yes

Reviewer #2: Yes

3. Has the statistical analysis been performed appropriately and rigorously? 

Reviewer #1: Yes

Reviewer #2: Yes

4. Have the authors made all data underlying the findings in their manuscript fully available?

Reviewer #1: Yes

Reviewer #2: Yes

5. Is the manuscript presented in an intelligible fashion and written in standard English?

Reviewer #1: Yes

Reviewer #2: Yes

6. Review Comments to the Author

Reviewer #1: (No Response)

Reviewer #2: (No Response)

7. PLOS authors have the option to publish the peer review history of their article (what does this mean? ). If published, this will include your full peer review and any attached files.

**Do you want your identity to be public for this peer review?** For information about this choice, including consent withdrawal, please see our Privacy Policy .

Reviewer #1: No

Reviewer #2: No

---

## [Editor Report · Acceptance letter]

PONE-D-25-13670R1

PLOS ONE

Dear Dr. Palmerio,

I'm pleased to inform you that your manuscript has been deemed suitable for publication in PLOS ONE. Congratulations! Your manuscript is now being handed over to our production team.

Kind regards,

on behalf of

Dr. Shoko Sugasawa

Academic Editor

PLOS ONE